# Unlocking the Antioxidant Potential of White Tea and Osmanthus Flower: A Novel Polyphenol Liquid Preparation and Its Impact on KM Mice and Their Offspring

**DOI:** 10.3390/foods12214041

**Published:** 2023-11-06

**Authors:** Yisen Wang, Jiaqi Xu, Ziluan Fan, Xun Zhou, Zhenyu Wang, Hua Zhang

**Affiliations:** 1School of Chemistry and Chemical Engineering, Harbin Institute of Technology, Harbin 150001, China; wangyisen20000518@163.com (Y.W.); xujiaqi_100fen@163.com (J.X.); wzy219001@163.com (Z.W.); 2School of Forestry, Northeast Forestry University, Harbin 150040, China; fzl_1122@163.com (Z.F.); 18846062296@163.com (X.Z.)

**Keywords:** liquid preparation, polyphenol, anti-fatigue, oxidative stress, white tea

## Abstract

White tea, known for its high polyphenol content, boasts impressive antioxidant properties, but its practical applications remain promising. In this study, we successfully developed a liquid polyphenolic preparation (wtofLPP) using white tea and osmanthus flowers, characterized by its rich antioxidant content and favorable rheological properties. This formulation offers a strong foundation for the creation and utilization of innovative antioxidant-rich food products. Notably, wtofLPP significantly enhanced the activity of certain antioxidant enzymes in both KM mice and their offspring, leading to a reduction in malondialdehyde (MDA) levels, prolonged swimming endurance, and a marked increase in levels of active antioxidant compounds. Furthermore, our study highlights that fatigue stress can impact offspring mice, suggesting that oxidative damage in parents may influence their offspring, potentially affecting their genetic function.

## 1. Introduction

Due to advancements in science and social progress, the demands of daily life and work have grown, leading to increased fatigue among individuals. Oxidative damage is a significant factor contributing to this fatigue, and it can have detrimental effects on the epigenetics of future generations. As a result, mitigating the effects of fatigue stress has become a prominent area of research [1,2,3]. In the 21st century, the healthcare industry emphasizes the concept of disease prevention rather than waiting for illnesses to manifest. The consumption of natural and nutritious foods plays a crucial role in supplementing essential antioxidants. Oxidative stress is strongly linked to bodily fatigue, and the excessive production of free radicals can have detrimental impacts on the epigenetics of the male reproductive system and its offspring. Antioxidants, on the other hand, can counteract the toxic effects of reactive oxygen and nitrogen free radicals, mitigating their harmful consequences [4,5].

Antioxidants can be categorized into two groups: chemically synthesized and naturally occurring. Chemically synthesized antioxidants have faced growing skepticism due to concerns about potential toxicity and carcinogenicity. On the other hand, natural antioxidants, derived from animals and plants, are favored for their potent antioxidant properties and high safety. Among these, plant-derived antioxidants are the most prevalent and widely recognized [6]. Polyphenols are compounds characterized by benzene rings and phenolic hydroxyl structures. They demonstrate a robust antioxidant capacity, surpassing that of vitamin E by approximately six to seven times. Additionally, polyphenols possess hypoglycemic and anti-aging properties, and they are abundantly found in various plants [7]. Researchers extract polyphenols from various natural sources, making use of their abundant active components and potent antioxidant properties [8]. These compounds are capable of efficiently neutralizing a substantial number of reactive oxygen species (ROS) and function as antioxidants, protecting against cellular damage [9].

Functional liquid preparations are becoming increasingly popular among consumers due to their health benefits and convenience. Traditional tea soaking methods may not cater to modern consumer needs, leading to the emergence of liquid tea preparations. White tea contains polyphenols and caffeine that can effectively eliminate excessive radicals in the body, delay aging, and enhance antioxidant capacity while promoting endocrine organ activity and reducing fatigue. Polyphenols of white tea have a remarkable antioxidant capacity; its hydroxyl scavenging ability is stronger than that of vitamin C, making it an excellent candidate for application in functional liquid preparations [10,11]. Functional tea preparations, due to their inherent antioxidant performance, have the potential to promote digestion, prevent colds, enhance immunity, and provide other health benefits. Therefore, developing white tea liquid preparations is vital for promoting the rapid growth of the whole tea preparation market.

This research paper aims to assess the impact of a liquid polyphenol preparation derived from white tea and osmanthus flower (wtofLPP) on antioxidant indices in KM mice and their offspring. The article seeks to illuminate the potential use of polyphenols as natural antioxidants for the prevention of diseases associated with oxidative stress.

## 2. Materials and Methods

### 2.1. Reagents and Materials

In this study, fresh Fuding white tea, harvested in 2021, was carefully selected and procured from Ningde City, Fujian Province. The tea was harvested within two years and stored in sterile packaging to maintain its freshness. Dried osmanthus flowers, also obtained in 2021, were purchased from Guangxi Zhuang Autonomous Region and transported in sterile packaging. Additionally, food-grade nicotinamide was procured from Jiangsu Brothers Pharmaceutical Co., Ltd. (Yancheng City, Jiangsu Province, China), while xanthan gum, CMC, and sodium alginate were sourced from Shandong Gaofeng Fermentation Co., LTD (Jinan City, Shandong Province, China). Food-grade taurine and L-carnitine were acquired from Hubei Yuanda Life Science and Technology Co., LTD (Wuhan City, Hubei Province, China).

### 2.2. Preparation of Liquid Formulations

The tea leaves were crushed using a grinder and subsequently extracted in an electric thermostatic water bath (HWS24, Shanghai Yiheng Technology Co., LTD., Shanghai Fengxian, China) at 80 °C for 45 min with a 1:80 (m:v) ratio [12]. Separately, osmanthus flowers were weighed and extracted in an electric thermostatic water bath (HWS24, Shanghai Yiheng Technology Co., LTD., Shanghai Fengxian) at 30 °C for 120 min using a 1:60 (m:v) ratio [13]. Following the extraction process, the samples were filtered at 100 mesh and centrifuged at 3000 rpm for 15 min, and the supernatants were taken as the extract. The extracts were concentrated at 50 °C [14] to half of the extraction volume. The two extracts were combined and homogenized using a 30 Mpa high-pressure homogenizer (Shandong Huaifang, Shandong Taihan Food Machinery Co., LTD, China) for 5 min [15]. The mixture was then sealed, sterilized at 63 °C for 30 min, and cooled at 4 °C for subsequent use.

The liquid formulations were optimized using the response surface methodology via Box–Behnken (4 factors and 3 levels) in Design-Expert 8.0.6 (Stat-Ease Inc, Minneapolis, MN, USA). The optimized formulation included a mixture of white tea soup, compound juice, and osmanthus extract in an 8:2:1 ratio with 10% added water. Niacinamide (4 mg/100 mL), 8% sugar, 0.2% compound stabilizer (xanthan gum:CMC:sodium alginate (2:3:1)) and taurine (50 mg/100 mL), L-carnitine (90 mg/100 mL), 0.025% potassium sorbate, and 0.03% sodium dehydroacetate were also added. Among these components, nicotinamide serves as a fundamental vitamin supplement, taurine functions as an essential amino acid supplement, and L-carnitine is employed as a nutritional supplement. The mixture (2.5 g/100 mL) was then homogenized twice at 30 MPa and sterilized as described above.

### 2.3. Particle Size and Rheological Performance

The average particle size of wtofLPP was determined using a nanoparticle analyzer. A 1 mL aliquot of the diluted sample was introduced into the preparation tank (medium: water) and maintained at room temperature. Subsequently, the sample was subjected to laser irradiation within the range of 0.3 nm to 10.0 μm [16] and recorded at 25 °C across a frequency range of 0.1 Hz to 100 Hz and a shear rate range of 0.1 to100 s^−^¹.

### 2.4. Antioxidant Performance In Vitro

The ABTS radical scavenging activity was determined by a previously reported method with slight modifications [17]. The absorbance A_1_ was obtained after 6 min of light protection at 25 °C. A parallel control experiment was conducted by replacing the wtofLPP with an equal volume of distilled water, yielding the reference absorbance (A_0_). The ABTS radical scavenging activity was calculated using the following equation:Clearance rate: M(%) = (A_0_ − A_1_)/A_0_ × 100%(1)

The superoxide radical scavenging ability was determined using the pyrogallol autoxidation method as described [18]. The wavelength was set at 320 nm, and the light absorption value A was measured every 30 s for 4 min. The absorption value or control absorbance A_0_ was determined by Tris-Hcl solution instead of sample. The clearance rate (Y) was calculated as follows:Clearance rate: Y(%) = (ΔA_0_ − ΔA)/ΔA_0_ × 100%(2)

The hydroxyl radical scavenging ability of the wtofLPP was assessed by following a modified protocol [19]. Absorbance was measured after reaction for 30 min.
Clearance rate: Z(%) = [A_0_ − (A_i_ − A_i0_)]/A_0_ × 100%(3)

The reducing power of the antioxidant wtofLPP was determined using a standard procedure [20].

To evaluate the synergistic resistance effect of each component of the wtofLPP and taurine on fatigue-induced oxidative stress, the Chou–Talalay model was employed for calculating the combination index (CI). It can be expressed using the following equation:
(4)$$\mathrm{Combination}\;\mathrm{Index}\;(\mathrm{CI}):\;\mathrm{CI}=\frac{D_{1}+D_{2}}{{\mathrm{DX}}_{1}+{\mathrm{DX}}_{2}}$$

Here, DX_1_ and DX_2_ denote the doses corresponding to the clearance rate (expressed as X%) when only D1 or D2 is present, respectively. Conversely, D_1_ and D_2_ represent the respective doses required to achieve the same clearance rate (X%) when the two substances are combined.

### 2.5. Animal Experiment

Sixty specific-pathogen-free (SPF) KM mice (22 ± 2 g, male, 4 weeks of age, SPF) were purchased from Liaoning Changsheng Biotechnology Co., LTD. (Shenyang, Liaoning, China). After a 7-day acclimatization period, the mice underwent a weight-bearing exhaustive swimming experiment on day 8. The exhaustive swimming experiment was employed to create a model of exercise-induced fatigue. They were randomly assigned to six groups: normal control (normal), model control (model), low-dose wtofLPP (LDG) (30 mL/kg), medium-dose wtofLPP(MDG) (60 mL/kg), high-dose wtofLPP (HDG) (90 mL/kg), and a positive control group (PCG) (China Beijing Red Bull Vitamin Beverage Co., LTD Red Bull Beverage 60 mL/kg). Following oral gavage, each group rested for 30 min. The mice were then subjected to a 30 min weight-bearing swimming training session, with their tails bound with lead skin (5% body weight) and submerged in a water tank maintained at a temperature of 25 ± 1 °C and a depth of 30 cm. Treatment groups received low, medium, or high doses of wtofLPPs once daily, while the normal control and model control groups were administered distilled water on an empty stomach for 28 consecutive days. During the exhaustive test, the exhaustive time was recorded when the mice were submerged at the bottom for more than 10 s.

All animal experimental procedures (Section 2.5 and Section 2.6) conformed with the Guidelines of the Institutional Animal Care Use Committee, Heilongjiang province, China. The animal protocols were approved by the Animal Ethics Committee of Harbin Institute of Technology (IACUC-2020032).

### 2.6. Anti-Fatigue in Offspring Mice

Sixty KM mice (male and female) with an average weight of 22 ± 2 g were selected. To ensure the consistency of the experimental animals, the mice were purchased from the same batch as the mice in Section 2.5 for the construction of offspring animal oxidative stress models and anti-fatigue experiments. The mice were divided into three groups: a normal group, a model group (treated with distilled water), and a wtofLPP administration group (treated with a medium dose of wtofLPP). Following 28 days of continuous gavage, mating was conducted in the pairs. After a week, the pregnant female rats were housed individually and allowed to give birth to offspring. At 28 days postnatal, the offspring mice were subjected to 24 h of food and water deprivation, after which blood samples were collected for analysis. Commercial kits were utilized to measure serum biochemical parameters, including antioxidant enzyme system indexes (SOD, CAT, and MDA), non-enzymatic antioxidant substance indexes (GSH, cupriplasmin, and vitamin E), and anti-fatigue indexes (BUN, LA, LDH, liver glycogen, and muscle glycogen).

### 2.7. Data Analysis

Data analysis was executed using SPSS 20.0 software (SPSS Inc., Chicago, IL, USA). All results were expressed as means ± standard deviation (SD). One-way analysis of variance (ANOVA) was performed, followed by Tukey’s test and Duncan’s multiple comparison tests to assess the statistical significance of differences between groups. A *p*-value of less than 0.05 (*p* < 0.05) was considered statistically significant [21].

## 3. Results

### 3.1. Performance of wtofLPP

The liquid polyphenol preparation is a fluid product primarily composed of white tea soup. Ensuring the stability of this preparation is of paramount importance, with rheological property indices and particle size analysis serving as crucial parameters for evaluating its stability [22].

Particle size analysis is a valuable method for evaluating the stability of liquid formulations [23]. In Figure 1a, it is evident that the particle size of the wtofLPP was 842 ± 46 nm, with a PDI of 0.435. These values indicate a homogeneous and well-dispersed formulation with a small particle size. The smaller the particle size, the higher the stability of the wtofLPP and the less prone it is to agglomerate into clusters.

Shear viscosity is defined as the rate of shear stress during steady flow, and its value can provide insights into the stability of aqueous solutions [24]. As depicted in Figure 1b, the viscosity of wtofLPP decreased significantly with increasing shear rate from 0.1 s^−1^ to 100 s^−1^. The elevated shear rate disrupted the structure of the wtofLPP particles, resulting in decreased binding between the molecules and a rapid decline in viscosity [25]. The most substantial reduction in viscosity was observed when the shear rate was 0.1 s^−1^~10 s^−1^. When the shear rate exceeded 10 s^−1^, the viscosity stabilized, and the shear force destroyed the flocculation between particles, resulting in a decrease in viscosity [25]. Ultimately, the molecular structure was almost destroyed, exhibiting pseudoplastic fluid characteristics. This property is beneficial for pipeline transportation during production and reduces energy consumption.

The energy storage modulus (G′) and loss modulus (G″) are the two most commonly used parameters to characterize the viscoelastic properties of substances [26]. The changes in the energy storage and energy dissipation moduli of the wtofLPP are illustrated in Figure 1c. Both the energy storage modulus and energy dissipation modulus exhibit a gradual increase within the frequency range of 0.1 Hz~100 Hz, indicating that wtofLPP exists in a viscoelastic state. Notably, the G′ values are higher than the G″ values in the system, which implies that the wtofLPP is predominantly elastic, with higher elastic behavior than viscous behavior.

### 3.2. Antioxidant Activity In Vitro

The antioxidant activity of the polyphenol liquid preparation was evaluated in vitro based on its excellent stability. The evaluation included a test for ABTS, superoxide radicals, hydroxyl radicals, and reducing power. To calculate the combination index (CI), the Chou–Talalay model was established, and the combined effect of taurine and principal component was analyzed.

As shown in Figure 2a, the scavenging ability of each component on ABTS free radicals continued to increase in the concentration range of 0–1.2 mg/mL, even reaching 66.73% when the concentration was 1.2 mg/mL, and the scavenging rate was higher than that of the ternary compound. The IC_50_ value of ABTS free radical scavenging of wtofLPP was calculated, and the CI index was found to be 0.94 ± 0.01, indicating that the ternary complex of wtofLPP has a synergistic antioxidant effect with taurine.

As shown in Figure 2b, the IC_50_ value of wtofLPP was 10.08, and the CI index was 0.94 ± 0.02, which was higher than that of other groups, indicating a synergistic effect among the components. The scavenging activity of wtofLPP on superoxide free radicals reached 68.77% when the concentration reached 12 mg/mL.

As shown in Figure 2c, the CI index of wtofLPP was 0.97 ± 0.01, indicating a synergistic effect among the components, but it was not significant. At a 12 mg/mL concentration, the hydroxyl radical scavenging rate of fruit juice was 72.11%, that of the white tea soup was 63.02%, and that of the osmanthus extract was 32.27%. The hydroxyl radical scavenging activity of wtofLPP reached 67.61%.

In Figure 2d, the absorbance was found to be stronger with greater antioxidant capacity, and it gradually increased with the concentration. At a concentration of 12 mg/mL, the absorbance of wtofLPP was 0.801, which was higher than that of the ternary compound (0.683), juice (0.787), tea soup (0.462) and osmanthus extract (0.372). This indicates that a synergistic antioxidant effect could be achieved among the components.

As an acidic free amino acid, taurine has the effect of relieving oxidative stress, inhibiting free radicals, and resisting oxidation. Osmanthus and juice also have certain antioxidant capacities, but the antioxidant capacity of tea soup is relatively better. It can be seen that after mixing, the antioxidant capacity of wtofLPP is further improved, which is significantly higher than that of the single component and ternary compound.

### 3.3. Effects on Weight-Bearing Swimming and Body Weight in Mice

The body weight of a rodent is indicative of its overall health status [27]. In this study, body weight measurements were taken every 4 days after a 7-day period of adaptive feeding and a subsequent 28-day intragastric administration of the liquid polyphenol preparation. The influence of wtofLPP on the body weight of mice is shown in Figure 3a. After the four-week intragastric administration, the body weight of mice in all groups, except for the normal group, increased gradually. Meanwhile, the body weight of swimming-trained mice was lower than that of the normal group. By the fourth week of the experiment, the body weight of the mice had stabilized, with the medium- and low-dose groups exhibiting the slowest rates of increase. The water intake of the mice in each group was normal, and there were no instances of mortality during the study.

The weight-bearing swimming test is a commonly used method to evaluate the exercise endurance of mice, as it correlates with their ability to withstand physical exertion. As shown in Figure 3b, single-factor variance analysis revealed that the mice’s exhaustion time in each administration group was significantly extended compared to that of the model group. Furthermore, within the dose range examined, the exhaustion time increased with higher doses of the polyphenol preparation, indicating that the wtofLPP can significantly improve the exercise endurance of mice.

### 3.4. Changes in Antioxidant Enzyme Indexes in Mice

SOD is a marker of protein damage that plays a critical role in maintaining the redox balance of the body by removing excess free radicals and catalyzing superoxide anions to dismutate [28]. As shown in Figure 4a, the SOD value in the model group was significantly lower than that of the normal group (*p* < 0.01), indicating that fatigue leads to redox imbalance and oxidative stress. In contrast, the SOD values of the medium- and high-dose groups were significantly higher than that of the model group (*p* < 0.01), suggesting that the wtofLPP can effectively enhance SOD activity in mice.

GSH-Px is a peroxidase that can inhibit free-radical reactions and catalyze the reaction of H_2_O_2_ with GSH to produce H_2_O_2_ reduction products [29]. As shown in Figure 4b, the GSH-Px activity in the model group was significantly lower than that of the normal group (*p* < 0.01). Notably, the GSH-Px activity of each wtofLPP group was higher than that of the model group, and the difference was significant (*p* < 0.01), which indicates that wtofLPP can effectively increase the GSH-Px activity in mice in a dose-dependent manner.

CAT is an important marker enzyme in the antioxidant enzyme system [30], which can convert H_2_O_2_ produced by SOD disproportionation into harmless water, and is an essential component of the biological defense system. As shown in Figure 4c, the CAT activity in the model group was significantly lower than that of the normal group (*p* < 0.01). The CAT activity in the high-dose group of the wtofLPP was significantly increased compared with that in the model group (*p* < 0.01), suggesting that the wtofLPP can effectively enhance CAT activity in mice.

MDA is a marker of lipid peroxidation, and its level reflects the degree of damage to various organs [31]. Excessive MDA leads to abnormal metabolism in the body. As shown in Figure 4d, the MDA content in the model group was significantly higher than that of the normal group (*p* < 0.01), indicating that fatigue-induced oxidative stress increased lipid peroxidation in mice. Notably, the MDA content of the wtofLPP groups was significantly lower than that of model group (*p* < 0.01), indicating that wtofLPPs can reduce the production of lipid peroxidation and protect against oxidative stress in mice [32].

### 3.5. Changes of Nonenzymatic Antioxidant Indices in Mice

The effect of wtofLPP on GSH synthesis in mice is shown in Figure 5a. The GSH level in the model group was significantly lower than in the normal group (*p* < 0.01), indicating its depletion in combating free radicals [33]. Compared to the model group, the GSH level in the wtofLPP groups increased significantly (*p* < 0.01) in a dose-dependent manner [34].

The effect of wtofLPP on the synthesis ability of ceruloplasmin in mice is shown in Figure 5b. The ceruloplasmin production in the model group was significantly lower than in the normal group (*p* < 0.01). In contrast, the production of ceruloplasmin in the wtofLPP groups was significantly higher than in the model group (*p* < 0.01).

Vitamin E can inhibit lipid peroxide production and reduce oxidative damage in the body [35,36]. The effects of wtofLPP on the synthesis ability of Vitamin E in mice are shown in Figure 5c. The vitamin E level in the model group was significantly lower than that in normal group (*p* < 0.01), while the high-dose wtofLPP group exhibited a significantly higher vitamin E level compared to the model group (*p* < 0.01).

### 3.6. Changes in Biochemical Parameters Related to Fatigue

Blood urea nitrogen (BUN) is an important indicator of protein degradation [37]. As shown in Figure 6a, the model group exhibited a significantly higher BUN level than the normal group, reaching the highest BUN level, suggesting that the body was experiencing an energy deficiency, leading to the production of a considerable amount of protein and amino acid metabolites [38]. The BUN levels in the wtofLPP groups were significantly lower than those in the model group (*p* < 0.01), displaying dose dependency. The wtofLPP was involved in the mice’s energy metabolism and significantly reduced their BUN levels.

Excessive LA can decrease exercise performance [39]. The effect of the wtofLPP on LA levels in mice is demonstrated in Figure 6b. The LA content in the model group was considerably higher than that in the normal group (*p* < 0.01). Compared with the model group, the LA content in the medium- and high-dose wtofLPP groups and positive control group were significantly decreased (*p* < 0.01), showing that polyphenols can significantly reduce lactic acid content, providing an anti-fatigue effect [40].

The effect of wtofLPP on the LDH level of mice is shown in Figure 6c). The serum LDH value in the model group was significantly decreased (*p* < 0.01). In contrast, the LDH levels in the wtofLPP groups were significantly higher than those in the model group (*p* < 0.01), indicating increased LDH levels following intragastric administration.

As shown in Figure 6d, the content of muscle glycogen in the model group was significantly lower than that in the normal group, and the content of muscle glycogen in wtofLPP at medium and high doses was significantly higher than that in the model group (*p* < 0.01), indicating that wtofLPP can delay fatigue.

### 3.7. Histopathology

Hematoxylin–eosin (HE) staining is a widely employed technique for pathological examination. Figure 7a demonstrates that the lung tissue in the normal group exhibited normal morphology, distinct cytoplasmic nucleolar staining, no fibrous hyperplasia in the lung interstitium, and normal bronchial mucosa at all levels. In contrast, the model group showed local alveoli dilation, mucous membrane shedding, the presence of blood cells in the lumen, inflammatory cell aggregation around the trachea, and an abundance of red blood cells in the alveolar epithelium. The lung tissue in the wtofLPP groups exhibited varying degrees of improvement.

Figure 7b reveals that the spleen tissue of the control group contained no macrophages and displayed a distinct boundary between white and red pulp without hyperplasia. In the model group, severe white pulp shrinkage and the presence of macrophages were observed, consistent with previously reported findings [41]. The high-dose group showed more significant improvement compared to the model group, and the treatment effect was dose-dependent.

As illustrated in Figure 7c, the myocardial fibers (normal group) were closely arranged with clear nuclei, no shrinkage, and an absence of blood cells. In the model group, the cardiac fiber arrangement exhibited widened gaps, numerous blood cells, and scattered inflammatory cells in the nucleus, which was consistent with the literature reports [41]. The myocardial pathological changes were reduced in all dose groups, with the most substantial improvement observed in the high-dose group.

Figure 7d indicates that the liver cell plates in the normal group were neatly and radially arranged, with distinct cytoplasmic and nuclear boundaries. Conversely, the model group displayed irregular arrangements, cytoplasmic fusion, swelling, and nuclear contraction. The liver tissue pathological changes were mitigated in all dose groups, with the high-dose group exhibiting the most significant improvement.

Figure 7e shows that the kidney tissue of the normal group appeared relatively normal, without obvious edema or inflammatory cell infiltration. In the model group, glomerular atrophy, epithelial exfoliation, conspicuous vascular congestion in the interstitium, and localized interstitial hyperplasia were observed. The pathological changes were improved across all dose groups.

As shown in Figure 7f, muscle tissues in the normal group were closely arranged with evenly stained cytoplasmic nuclei and no edema. The model group exhibited an irregular muscle tissue arrangement, prominent edema, and nucleolar shrinkage, which was consistent with previously published findings [42]. The high-dose wtofLPP group demonstrated better improvement compared to the low- and medium-dose groups.

### 3.8. Effects on Mice Fertility and Offspring Weight

The average number of offspring produced by each mouse in each group is depicted in Figure 8a. No significant difference in offspring numbers was observed among the wtofLPP groups (LP-LP, N-F, and N-M) when compared to the normal group. The fertility rate of the model group, however, exhibited a significant decrease, indicating that the wtofLPP could protect against fatigue-induced parental fertility decline. The fertility rates of the N-F and M-F groups were lower than those of the M-M and N-M groups, implying that oxidative damage caused by fatigue in female mice had a more pronounced impact on fertility.

As shown in Figure 8b, in general, the weight of the male offspring mice in each group was higher than that of the female mice. When compared to the normal group (male 18.62 ± 1.23 g, female 16.86 ± 0.91 g), the weight of the male offspring mice in the other groups decreased, with the model group (male 16.42 ± 1.30 g, female 15.34 ± 1.59 g) experiencing the most significant weight loss, while the other groups had more moderate weight reductions. Compared with the normal group, the model males lost 13.40% of their body weight and the females lost 9.9% of their body weight, which indicates the greater effect of fatigue induction in male mice.

### 3.9. Effects on Antioxidant of Mouse Offspring

Figure 9a illustrates that SOD activity in each experimental group was significantly lower than that in the normal group (*p* < 0.01), with the lowest SOD activity observed in the model group. This indicates a considerable impact of fatigue on SOD activity. Notably, the activity of SOD in the N-F group was significantly higher than that in the model group (*p* < 0.05).

According to Figure 9b, the CAT activity in the model group was significantly reduced compared to that in the normal group. In contrast, the CAT activity significantly increased in the M-M and N-F groups treated with the wtofLPP (*p* < 0.05).

As depicted in Figure 9c, the MDA content in all groups increased compared to that in the normal group, suggesting that fatigue-induced oxidative damage affected the offspring. Compared to the model group, MDA content significantly decreased in the M-M, N-M, LP-LP, and N-F groups treated with the wtofLPP (*p* < 0.01). Compared to the M-F group, The MDA content of the offspring in the M-M and N-F groups demonstrated less difference comparing with the normal group (*p* < 0.01). In contrast, vitamin E levels in the N-F group treated with wtofLPP significantly increased (*p* < 0.01), with M-M and LP-LP groups also showing a significant increase (*p* < 0.05). The vitamin E levels of offspring in the M-M group were less different from the normal group when compared to the M-F group. Furthermore, vitamin E levels of offspring in the N-F group were closer to those in the normal group than in the N-M group, indicating that the oxidative damage in male mice had less impact on offspring than in fatigue-damaged female mice.

As shown in Figure 9e, ceruloplasmin levels in the model group were significantly lower than those in the normal group (*p* < 0.01). The ceruloplasmin levels in the N-F and M-M groups treated with the wtofLPP significantly increased (*p* < 0.01), and the levels of ceruloplasmin in the N-M and LP-LP groups also significantly increased (*p* < 0.05). Compared to those in the M-F group, the ceruloplasmin levels of the offspring in the M-M and N-F groups were less different from the normal group, indicating that fatigue-induced oxidative damage had a lesser effect on offspring in male mice than in the normal group.

Lastly, Figure 9f demonstrates that the GSH synthesis ability in the model group was significantly lower than that in the normal group (*p* < 0.01). The synthesis ability of GSH in the N-F and M-M groups was significantly higher than that in model group (*p* < 0.01), and their GSH synthesis ability improved to some extent.

### 3.10. Effects on Fatigue Resistance of Mouse Offspring

As shown in Figure 10a, the BUN level in the model group was significantly higher compared to that in the normal group, indicating increased fatigue. Conversely, the BUN levels in the N-F, M-M, and N-M groups treated with wtofLPP were significantly lower than those in the model group (*p* < 0.01). The LP-LP group also exhibited a substantially reduced BUN level compared to the model group (*p* < 0.05). These results suggest that the wtofLPP can reduce exercise-induced protein catabolism and nitrogen excretion in offspring.

Figure 10b shows that the LA level in the model group was notably higher than that in the normal group (*p* < 0.01). The N-F group treated with polyphenol liquid preparation displayed a significantly lower LA level than the model group (*p* < 0.05), while no significant differences were observed in other groups (*p* < 0.05). These findings suggest that the experimental administration groups can enhance lactic acid aerobic metabolism, alleviating fatigue [43]. The M-M and N-F groups exhibited a smaller difference in offspring LA levels compared to the M-F and N-M groups, indicating that fatigue-induced oxidative damage in female mice had a more pronounced impact on offspring than that in male mice.

As depicted in Figure 10c, liver glycogen content was substantially reduced in the model group, while the content of liver glycogen in the LP-LP, N-F, M-M, and N-M groups treated with wtofLPPs displayed significantly higher liver glycogen levels than the model group (*p* < 0.01). This suggests that the wtofLPPs can mitigate liver glycogen consumption in vivo. Exercise-induced physical exhaustion occurs concurrently with liver glycogen depletion, as liver glycogen is progressively broken down to maintain the blood glucose balance and support exercise, leading to a decline in liver glycogen levels.

Lastly, Figure 10d reveals that muscle glycogen content in the model group was significantly reduced (*p* < 0.01). The M-M group treated with wtofLPP exhibited a substantially higher muscle glycogen content than the model group (*p* < 0.01), while the N-M group showed a significant increase (*p* < 0.05). No significant differences were observed among the other groups. These results indicate that increased glycogen reserves can delay the onset of fatigue [44].

## 4. Discussion

In the study of antioxidant properties of wtofLPPs in vitro, a comparative analysis was conducted by SPSS software to compare the IC_50_ values. In addition to this, the combined action index was also calculated. The result revealed that the combination of each component of wtofLPPs and taurine had a synergistic effect (Figure 2), indicating that the various components of wtofLPP have a good synergistic effect and that the combination of liquid preparation is reasonable.

In vivo experiments with KM mice revealed that wtofLPP offered protection against oxidative stress in parent mice, effectively enhancing the activities of GSH-Px, SOD, and CAT in mice and their offspring. It also reduced MDA levels; improved the synthesis capabilities of vitamin E, GSH and ceruloplasmin; decreased inflammatory response; and mitigated the detrimental effects of oxidative damage (Figure 4, Figure 5 and Figure 6). These results indicated that wtofLPPs can maintain the stability of the antioxidant system in mice and their progeny. Exhaustive swimming resulted in changes to the physiological and biochemical indexes of the mice. Compared to the model group, the exhaustive time of mice in each administration group was prolonged, while BUN and LA levels decreased and glycogen contents increased. Morphological observation showed that wtofLPPs ameliorated the pathological changes in the heart, lung, liver, kidney, muscle, and spleen to varying degrees (Figure 7).

In the study of anti-fatigue in offspring mice, the intragastric administration of wtofLPP effectively decreased serum LA and BUN concentrations and increased glycogen levels in offspring, but the difference was not significant. Additionally, parental administration of the preparation enhanced the offspring’s production and improved the fertility rate of fatigue-damaged mice, with a more pronounced effect on female mice (Figure 8, Figure 9 and Figure 10). Nevertheless, there were no significant differences in the sex ratio and body weight of the offspring. It is worth noting that oxidative damage appears to affect males more significantly than females.

All the results mentioned above suggest that wtofLPP can enhance the endurance of mice with oxidative damage, reduce the accumulation of fatigue-related metabolites, and provide protection against oxidative damage caused by fatigue. This indicates that wtofLPP possesses significant antioxidative properties and, being non-toxic, holds certain practical value for application.

## 5. Conclusions

In summary, this study highlights the favorable taste and stability of wtofLPPs. Through in vivo experiments, it has been demonstrated that wtofLPP can mitigate the accumulation of metabolites resulting from fatigue, enhance the body’s capacity to synthesize antioxidant enzymes and non-enzymatic antioxidant substances, safeguard against damage induced by fatigue stress, and postpone the onset of fatigue. Furthermore, the study reveals that fatigue stress can affect offspring mice, suggesting that oxidative damage in parents may impact the offspring, possibly even their genetic function. These findings establish a robust theoretical basis for understanding the pathways of oxidative damage and their implications for fertility. Moreover, wtofLPP provides a theoretical foundation for the development and application of new antioxidant-rich foods and an expectation of having a strong antioxidant stress effect in fields such as military traumatic stress, sports training, and high-intensity work.

## Figures and Tables

**Figure 1 foods-12-04041-f001:**
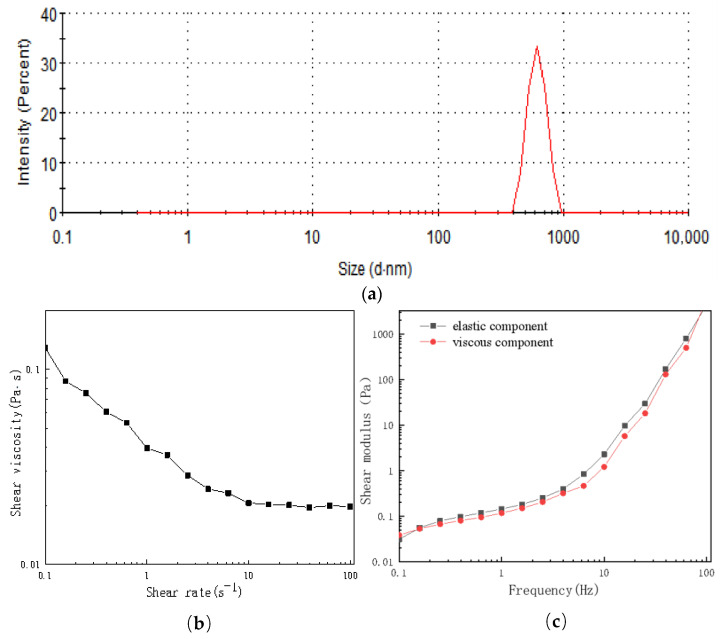
(**a**) Particle size analysis of wtofLPPs. (**b**) Effect of shear rate on viscosity of wtofLPPs. (**c**) Changes in storage modulus and energy.

**Figure 2 foods-12-04041-f002:**
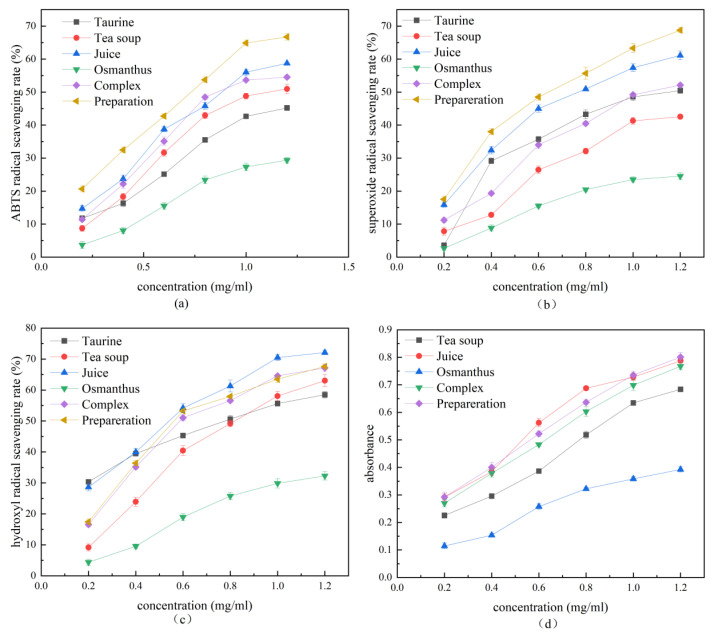
(**a**) ABTS radical scavenging rate; (**b**) superoxide radical scavenging rate; (**c**) hydroxyl radical scavenging rate; (**d**) absorbance.

**Figure 3 foods-12-04041-f003:**
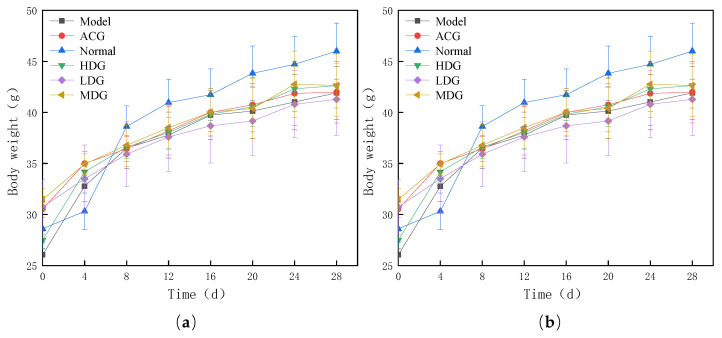
(**a**) Effects of wtofLPP on body weight of mice. (**b**) Effects of wtofLPP on exhaustion time of mice. Note: Model: model group; ACG: positive control group; Normal: normal group; LDG: low-dose group; MDG: medium-dose group; HDG: high-dose group, same figure below.

**Figure 4 foods-12-04041-f004:**
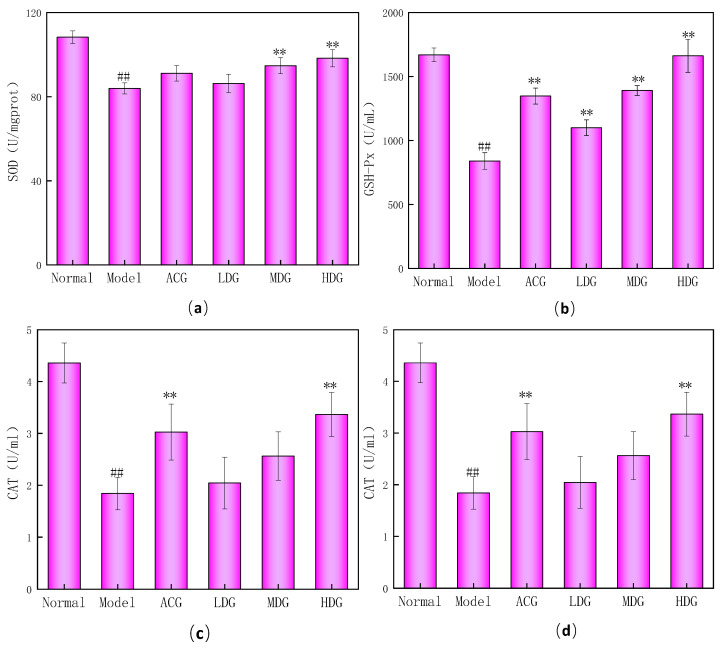
(**a**) Effects of wtofLPP on SOD in mice. (**b**) Effects of wtofLPP on GSH-Px in mice. (**c**) Effects of wtofLPP on CAT in mice. (**d**) Effects of wtofLPP on MDA in mice. Note: ##: extremely significant difference compared with normal group, *p* < 0.01; **: extremely significant difference compared with model group, *p* < 0.01. Same figure below.

**Figure 5 foods-12-04041-f005:**
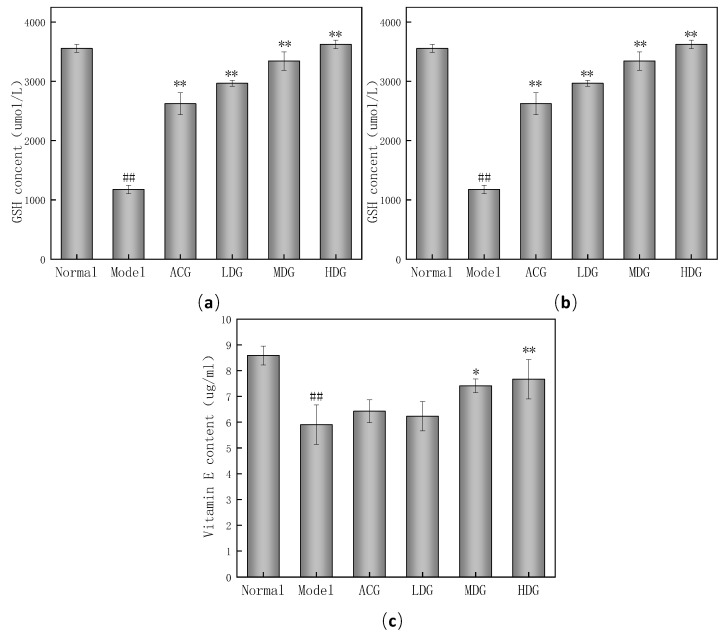
(**a**) Effects of wtofLPP on the synthesis of GSH in mice. (**b**) Effects of wtofLPP on the synthesis of ceruloplasmin in mice. (**c**) Effects of wtofLPP on the synthesis of vitamin E in mice. Note: ##: extremely significant difference compared with normal group, *p* < 0.01; *: significant difference compared with model group, *p* < 0.05; **: extremely significant difference compared with model group, *p* < 0.01.

**Figure 6 foods-12-04041-f006:**
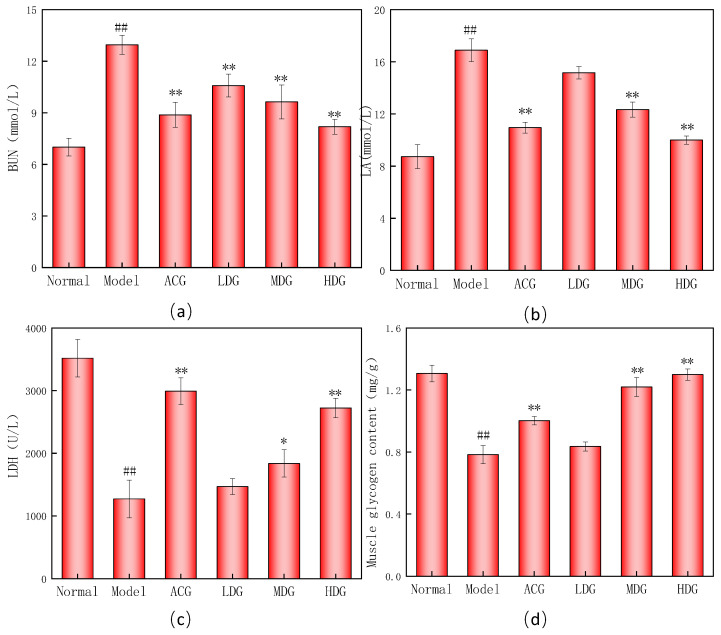
(**a**) Effects of wtofLPP on BUN level of mice. (**b**) Effects of wtofLPP on LA level of mice. (**c**) Effects of wtofLPP on LDH level of mice. (**d**) Effects of wtofLPP on muscle glycogen level of mice.Note: ##: extremely significant difference compared with normal group, *p* < 0.01; *: significant difference compared with model group, *p* < 0.05; **: extremely significant difference compared with model group, *p* < 0.01.

**Figure 7 foods-12-04041-f007:**
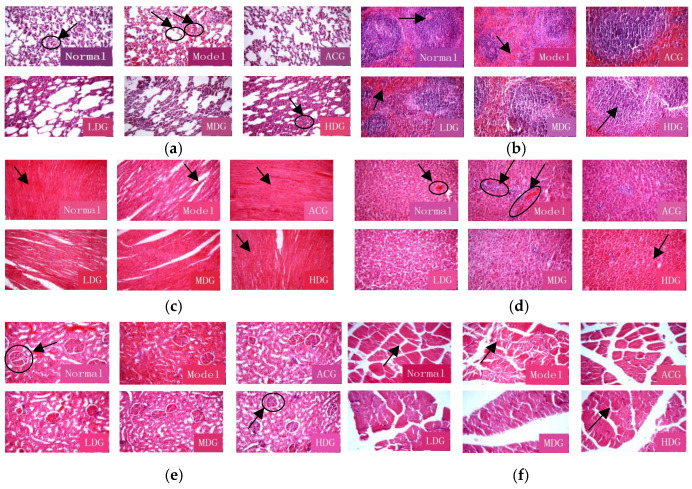
(**a**) Influence of lung histomorphology. (**b**) Influence of spleen histomorphology. (**c**) Influence of heart histomorphology. (**d**) Influence of liver histomorphology. (**e**) Influence of kidney histomorphology. (**f**) Influence of muscle histomorphology.

**Figure 8 foods-12-04041-f008:**
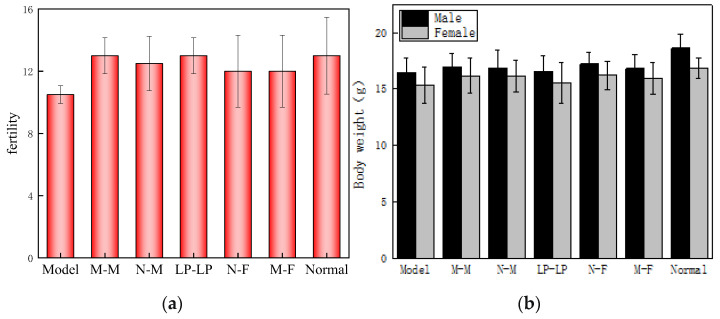
(**a**) Parental fertility rate. (**b**) Changes of offspring body weight in each group. Note: Model: model male–model female; M-M: model male–normal female; N-M: normal male–liquid preparation female; LP-LP: liquid preparation male–liquid preparation female; N-F: preparation male–normal female; M-F: normal male–model female; Normal: normal male–normal female; the following pictures are the same.

**Figure 9 foods-12-04041-f009:**
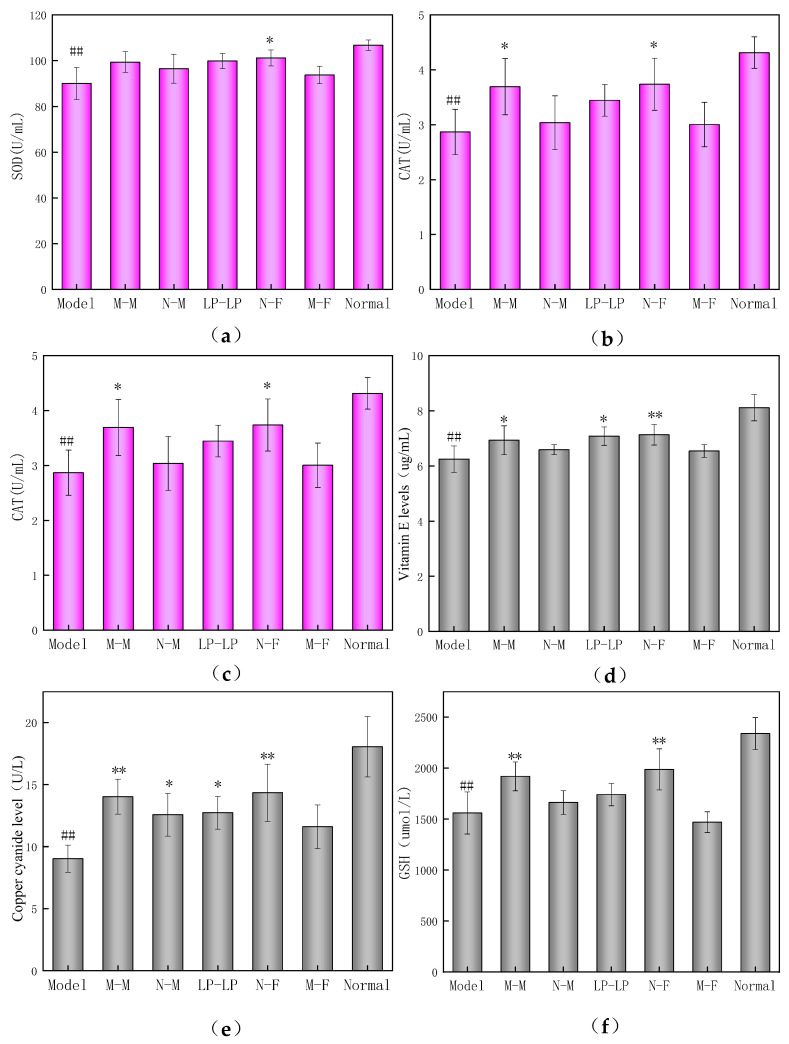
(**a**) Changes in SOD activity of offspring. (**b**) Changes in CAT activity of offspring. (**c**) Changes in MDA content. (**d**) Changes in VE level of offspring. (**e**) Changes in ceruloplasmin level of offspring. (**f**) Changes in GSH synthesis ability of offspring. Note: ##: extremely significant difference compared with normal group, *p* < 0.01; *: significant difference compared with model group, *p* < 0.05; **: extremely significant difference compared with model group, *p* < 0.01.

**Figure 10 foods-12-04041-f010:**
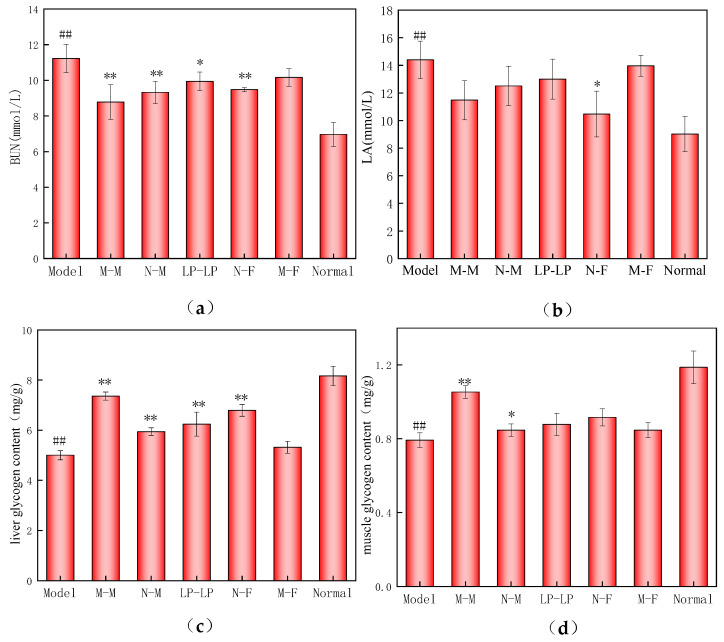
(**a**) Changes in BUN level of offspring; (**b**) changes in LA level of offspring; (**c**) changes in liver glycogen of offspring; (**d**) changes in muscle glycogen of offspring. Note: ##: extremely significant difference compared with normal group, *p* < 0.01; *: significant difference compared with model group, *p* < 0.05; **: extremely significant difference compared with model group, *p* < 0.01.

## Data Availability

Data are contained within the article.

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
