# Peer review of "Unlocking the Antioxidant Potential of White Tea and Osmanthus Flower: A Novel Polyphenol Liquid Preparation and Its Impact on KM Mice and Their Offspring"

_foods, 2023, doi:10.3390/foods12214041_

Round 1
Reviewer 1 Report
Comments and Suggestions for Authors
The article titled "Unlocking the Antioxidant Potential of White Tea and Osmanthus Flower: A Novel Polyphenol Liquid Preparation and Its Impact on KM Mice and Their Offspring" presents research on a new liquid preparation's antioxidant potential. Below are points of improvement for this study:
Line 10: Replace "but there is still some space" with "yet there remains potential". Line 12-13: The information about the nanoparticle analyzer seems out of place. Consider moving it to the methodology section. Line 14-22: The results and findings are slightly cluttered. Streamline them to ensure the reader can quickly understand the primary outcomes. Line 26-27: The introduction starts by immediately diving into the specifics. It would be more effective to begin with a general introduction about the importance of antioxidants before detailing the study's aim. Line 28-34: Some information is repetitive from the abstract. Try to provide more context rather than restating the research aim. Line 36-40: This section on oxidative stress could be more concise. Line 48-64: This is more of a background on tea preparations. It could be summarized more effectively to provide a more concise introduction. Line 54: "Polyphenols, one of the active substances in tea," this seems redundant as polyphenols were already introduced earlier. Lines 65-78: These generic instructions about data availability, protocols, etc. do not seem necessary for this article. They appear to be instructions for the authors rather than content for the readers. Lines 82-102: The extraction methods are explained well, but the sequence could be more streamlined. Line 96-98: The added ingredients and their quantities are just listed. It would be helpful to explain the purpose of each additive briefly. Line 104-108: The description of using the nanoparticle analyzer seems abrupt. More context should be provided about why this was necessary. Line 134-143: For the animal experiments, the motivation behind the specific methods, such as weight-bearing swimming, could be elaborated more. Line 152-162: There seems to be a switch from talking about rats to talking about mice. Ensure consistency in the animals discussed. Line 164-167: The ethical statement is good but could be integrated with the section discussing animal experiments to give it better context. Lines 168-172: This could be a subsection called "Data Analysis" instead of being combined with "Statistical Analysis." Ensure consistency in naming conventions (e.g., "wtofLPP" vs. "WTOFLPP"). Introduce abbreviations upon their first appearance. For instance, the first mention of superoxide dismutase should include (SOD).
The article uses a lot of numerical data without appropriate context. It would be beneficial to provide a comparative analysis for easier interpretation of the results. More detail on the control groups used in the experiment could provide additional clarity. While there are mentions of figures, providing charts, or more illustrative representations of data might enhance understanding. Also, consider employing data visualization techniques that clearly differentiate between multiple groups. The use of abbreviations (e.g., wtofLPP, SOD, CAT) should be consistent throughout. Ensure each abbreviation is defined the first time it's introduced. Expand on the interpretation of results. For instance, why might wtofLPP have the effects observed? How does it compare to other treatments or control groups? The article could benefit from comparison with similar studies in the field. Are the findings in line with what has been previously observed or are there notable differences? Discussion on the potential mechanisms through which wtofLPP exerts its antioxidant and other protective effects would add depth to the study.
In conclusion, summarize the primary findings of the research in a concise manner, so that readers can quickly grasp the main takeaways. Discuss the broader implications of the findings. For example, if wtofLPP is found to be beneficial, how might it be applied in real-world scenarios? What could this mean for the future of antioxidant treatments? Suggest potential areas of research that could be explored based on the results of this study. The conclusion should be written in a way that even those not familiar with the technical details can understand the importance and relevance of the research.
Comments on the Quality of English LanguageModerate editing of English language required
Reviewer 2 Report
Comments and Suggestions for Authors
Find below my comments to improve the article
Abstract
Also, talk more about the critical findings. Results should cover 60% of the abstract.
I suggest you first draft it like this
Background (2 to 3 sentences)
Scope and approach (2 to 4 sentences)
Key findings
This should cover >60% of the abstract
Conclusions (2 to 3 sentences)
After you draft it like this, then you remove the sections (background, scope & approach, key findings and conclusions" and you merge all the sentences together.
Keywords
Add white tea
Line 23
Provide list of abbreviation after keywords
Lines 26 to 34
Is that how we start an introduction to a research paper? Read quality research papers and rewrite it again
Line 28
Define wtofLPP for the first time. Correct them in the entire manuscript.
Line 50
Also, talk about various remedies used to counteract ROS and RNS and their limitations. That will lead to the need to utilize polyphenols.
Line 64
It is better to move lines 26 to 34 to line 64. also, what is the research gap? Justify the novelty of this research.
Line 67 to 80
What is the need for these? Are you people serious about peer review? Delete them because this is very basic.
Line 83,
Talk about where the tea was harvested (longitude and magnitude) and the date. How was it transported and under what conditions?
Line 86
Why select these conditions for extraction? Provide reference for that.
Line 93
So no centrifugation and filtration?
Line 105
What is the concentration of the diluted sample?
Lines 177 to 178
Where is the reference for that? Same as lines 188, 193, etc. talk about the science behind them. Correct all in the entire manuscript
Line 198
Talk about the difference in antioxidant activity amongst various samples and the science behind this
Line 261
What do the star signs in figure 4 to 6 mean? Elaborate them underneath the figure.
Line 307
Figure 7. Put arrows in each figure to show what you mean. Also, critically discuss that section and talk about the science behind your results
Figure 8
Run statistical analysis on this figure and critically discuss them.
Line 436
The discussion section needs to be comprehensive. Compare and contrast the results with various literatures and talk critically about the science behind your results
Line 462
Rewrite the conclusion again. Critically summarize the key findings and talk about future research that needs to be done.
Comments on the Quality of English Language
moderate
Round 2
Reviewer 1 Report
Comments and Suggestions for Authors
Accept after minor revision
Comments on the Quality of English LanguageMinor editing of English language required
Reviewer 2 Report
Comments and Suggestions for Authors
I recommend the acceptance of this paper.
Comments on the Quality of English Languageminor